# Interdisciplinary Treatment of Malignant Chest Wall Tumors

**DOI:** 10.3390/jpm13091405

**Published:** 2023-09-20

**Authors:** Koblandy Khamitov, Wojciech Dudek, Andreas Arkudas, Mohamed Haj Khalaf, Chirag Parjiea, Mostafa Higaze, Raymund E. Horch, Horia Sirbu

**Affiliations:** 1Faculty of Medicine, Friedrich-Alexander-University Erlangen-Nürnberg, 91054 Erlangen, Germany; 2Department of Thoracic Surgery, University Hospital Erlangen, Krankenhausstr. 12, 91054 Erlangen, Germany; 3Department of Plastic and Hand Surgery, University Hospital Erlangen, 91054 Erlangen, Germany

**Keywords:** Special Issues, thoracic surgery, plastic surgery, reconstructive surgery, chest wall tumors, chest wall resection, chest wall reconstruction

## Abstract

Background: Chest wall resections for malignant chest wall tumors (MCWTs), particularly those with full-thickness chest wall involvement requiring reconstruction, present a therapeutic challenge for thoracic and plastic reconstructive surgeons. The purpose of this study was to review our experience with chest wall resection for primary and metastatic MCWTs, with a focus on perioperative outcomes and postoperative overall survival (OS). Methods: All patients who underwent surgical resection for primary and secondary MCWTs at our single institution between 2000 and 2019 were retrospectively analyzed. Results: A total of 42 patients (25 male, median age 60 years) operated upon with curative (n = 37, 88.1%) or palliative (n = 5, 11.9%) intent were reviewed. Some 33 (78%) MCWTs were of secondary origin. Chest wall reconstruction was required in 40 (95%) cases. A total of 13 (31%) patients had postoperative complications and one (2.3%) died perioperatively. The 5-year postoperative overall survival rate was 51.9%. The postoperative 5-year survival rate of 42.6% in patients with secondary MCWTs was significantly lower compared to the figure of 87.5% in patients with primary MCWTs. Conclusions: In well-selected patients, chest wall resections for primary and secondary MCWTs are feasible and associated with good perioperative outcomes. For secondary MCWTs, surgery can also be performed with palliative intent.

## 1. Introduction

Chest wall resections and reconstructions for malignant chest wall tumors (MCWTs) are challenging for thoracic and plastic reconstructive surgeons, especially in cases of full-thickness chest wall involvement. The extent of the chest wall resection must follow oncological principles, but on the other hand, it is important to preserve the bony stability of the chest [1]. Accurate patient selection, planning of surgery, and appropriate perioperative management are of particular importance. MCWTs requiring surgery can be subdivided into primary and secondary MCWTs. Primary MCWTs include tumors arising from soft tissues as well as cartilage and bone; these are most commonly sarcomas, and less commonly solitary plasmacytomas or lymphomas [2]. Secondary MCWTs may arise from the direct spread of breast or lung cancer or from distant spread of neoplasms from other sites in the body; their incidence is higher compared with primary tumors [3]. MCWT resections can be performed with curative or palliative intent. In the curative setting, the aim of surgical treatment is the resection of the tumor with microscopically negative margins. Palliative resections of painful, malodorous, and bleeding lesions with subsequent thoracic wall reconstruction aim to improve the patient’s quality of life [4]. This study aims to analyze the outcomes of multidisciplinary treatment of primary and metastatic MCWTs and to assess postoperative overall survival (OS).

## 2. Materials and Methods

### 2.1. Study Groups

All adult patients who underwent oncological resections for primary and secondary MCWTs at our single institution with curative or palliative intent between 2000 and 2019 were retrospectively analyzed in IBM SPSS Statistics Version 28 (IBM, Armonk, NY, USA).

### 2.2. Perioperative Assessment

Cardiorespiratory fitness was routinely assessed in all surgical candidates. The pre-operative assessment included physical examination, chest X-ray and computed tomography (CT); a positron emission tomography (PET) scan was performed to exclude disseminated metastatic disease. Surgical data such as the extent of resection, type of prosthetic material used for reconstruction, and type of flap used for the coverage of the chest wall defect were reviewed. Tumor histology, completeness of resection (R-status) and perioperative and long-term outcomes were assessed.

The standardized Clavien–Dindo classification was used to estimate postoperative complications [5]. Postoperative survival was calculated using a Kaplan–Meier survival analysis.

The need for induction or postoperative oncological therapy was discussed and planned within the institutional oncological multidisciplinary team conference.

### 2.3. Surgical Principle

Complete (R0) tumor resection was a key surgical principle. However, the intraoperative finding of more advanced tumor invasion compared to the carefully performed preoperative imaging was not an indication to discontinue surgery. Therefore, a specific safety margin for resection has not been defined. Rib, sternum and clavicle involvement was not considered a surgical contraindication. If extended soft-tissue or bone resection was required, interdisciplinary preoperative planning of the chest-wall defect’s reconstruction with the involvement of a plastic reconstructive surgeon took place. Especially in the case of large defects (>5 cm, >10 cm posterior or >2 resected ribs), prosthetic materials and muscle flaps were planned for chest wall reconstruction. The latissimus dorsi (LD) muscle flap was used to cover the anterior chest wall, including the sternal region. For reconstruction of the chest wall after sternotomy or after MCTW resection in the contralateral region, the pectoralis major (PM) flap was considered. For longitudinal anterior and anterolateral chest wall defects, coverage with a vertical rectus abdominis muscle flap (VRAM) or transverse rectus abdominis muscle flap (TRAM) was planned. Polyglactin and polypropylene meshes were frequently used as prosthetic materials due to their flexibility and plasticity. The permeability of these meshes helps to reduce postoperative complications such as seroma.

## 3. Results

A total of 42 patients (25 male) were analyzed. The median age was 60 years (range 25–86 years). Patient characteristics are shown in Table 1.

The majority of MCWTs were of metastatic origin (n = 33, 78.6%). The most common primary MCWTs were sarcomas (n = 8, 19.04%), followed by plasmacytoma (n = 1, 2.38%). Secondary MCWTs included eight patients (19.04%) with breast cancer metastases and two patients (4.76%) who had direct invasion from breast cancer. Six chest wall resections (14.29%) were performed for non-small cell lung cancer, four (9.52%) for metastatic renal cell cancer, 3 (7.14%) for malignant melanoma, and two (4.76%) for metastatic liposarcoma. Five other patients presented with metastatic colorectal cancer, hepatocellular carcinoma, prostate cancer, thyroid cancer and cervical cancer. Two patients had metastases of hemangiopericytoma and histiocytoma.

Thirty-seven chest wall resections (88.1%) were performed with curative intent. Five resections (11.9%) were of palliative intent, including two (4.76%) patients with ulcerating sternal malignant melanoma metastasis, one (2.38%) painful rib metastasis of cholangiocellular carcinoma, one (2.38%) necrotic cervical cancer metastasis, and one (2.38%) necrotic metastasis of cholangiocellular carcinoma. Detailed characteristics of the resections with palliative intent are shown in Table 2.

Eleven patients (26.1%) received neoadjuvant therapy, of which nine patients underwent radio-chemotherapy, one patient chemotherapy, and one (2.38%) patient radiotherapy. Detailed characteristics of the neoadjuvant therapy are shown in Table 3.

Thirteen (31%) patients had postoperative complications. Major (grade III-V) complications occurred in ten (23.8%) patients, including three with hemothorax and two with pleural empyema requiring surgical intervention. One patient required surgical hemostasis for acute muscle flap bleeding. Other complications included wound hematoma, wound dehiscence, and pleural effusion requiring a surgery. There was one perioperative death on the 24th postoperative day due to respiratory failure as a result of postoperative pneumonia; the 30-day mortality rate was 2.3%. Detailed characteristics of postoperative complications are shown in Table 4.

### 3.1. Extent of Resection

Thirty-seven (88.06%) patients required rib resection (parasternal n = 12 (28.56%), anterior n = 7 (16.66%), anterolateral n = 4 (9.52%), lateral n = 13 (30.94%), and dorsolateral n = 1 (2.38%)). Complete sternal resection was performed in 2 (4.76%) cases (Figure 1), and partial sternal resection in 14 (33%) cases (Figure 2), including 8 (19%) patients requiring manubrium resection. In four (9.5%) cases, resection of the sternoclavicular joint was performed (Figure 3). In 23 (54%) patients, the involvement of a plastic surgeon was required.

Tumor resection was complete (R0) in 30 (71.4%) patients. In four cases (9.5%), resection margins were intraoperative negative (R0); meanwhile, the final pathological examination gave positive margins (R1/2). Intraoperative findings were that four (9.5%) other patients were more pronounced than expected upon preoperative imaging. Therefore, despite intraoperative macroscopic complete (R0) tumor resection, the final pathological examination reported positive margins (R1/2). Furthermore, in one case (2.38%), complete (R0) resection was not possible due to extensive adhesions as a result of previous surgeries and radiation. In these nine (21.42%) cases, local control via radiotherapy was required.

In one case (2.38%) with a positive margin (R1/2) after chest wall resection for metastatic breast cancer, a repeated surgery was required. After the reoperation, the resection margin was negative (R0). After chest wall resection for advanced lung cancer with a positive margin (R1/2), one patient (2.38%) died due to respiratory failure due to postoperative pneumonia. In two other (4.76%) cases, complete (R0) tumor resection was not intended, as the goal of the surgery was palliative.

### 3.2. Chest Wall Reconstruction

Chest wall reconstruction was required in 40 (95%) cases. Prosthetic material and muscle flap were used in 22 (52.4%) cases, prosthetic material only in 14 (33.3%), muscle flap only in 4 (9.5%). In the majority of patients (n = 17, 40.4%), Prolene-Mesh (polypropylene) was used to cover the chest wall defect (Figure 2), followed by Marlex-Mesh (polyethylene and polypropylene) (n = 7, 16.7%), Vicryl-Mesh (polyglactin) (n = 5, 11.9%) and Parietex-Mesh (polyester with collagen film) (n = 2, 4.76%). Reconstruction with Dacron-Mesh (polyester) was performed in one case (2.38%), and with Gore-Tex-Mesh (PTFE) in one (2.38%) case. Chest wall reconstruction was also performed using osteosynthesis systems/titanium plates (Figure 1 and Figure 3). The distribution of muscle flaps and prosthetic materials used for the reconstruction is shown in Figure 4, Table 5, and Table 6, respectively.

### 3.3. Postoperative Survival

In order to determine the survival rate after chest wall resections for the MCWTs, cases with palliative intent (n—5, 11.9%) were excluded. Thus, the 5-year postoperative overall survival rate was 51.9% (Figure 5). The postoperative 5-year survival rate of 42.6% in 28 patients with secondary MCWTs who underwent surgery with curative intent was significantly lower compared to the value of 87.5% for 9 patients with primary MCWTs (Figure 6). For the two large secondary MCWTs subgroups in our series, the postoperative 5-year overall survival was 41.7% for patients who underwent chest wall resection for lung cancer and 53.3% for breast cancer. Tumor resection was completed (R0) in 30 (69.04%) patients. Paradoxically, the 5-year survival rate of patients with incomplete (R1/2) resections was higher (77.1%) than that of patients with complete resection (42.4%, Figure 7).

### 3.4. Adjuvant Therapy

Some 19 (45.23%) patients, including 3 (7.14%) with primary MCWTs, required adjuvant therapy. Eight (19.04%) received radio-chemotherapy, seven (16.6%) chemotherapy, three (7.14%) radiotherapy, and one (2.38%) hormone therapy. The detailed characteristics of the adjuvant therapy are shown in Table 7.

## 4. Discussion

According to the largest reports, 40% of chest wall surgery is performed for directly invasive lung cancer, 10% to 20% for metastatic breast cancer, and 30% for primary chest wall tumors [6,7,8]. Approximately half of primary CWTs are benign [9]. MCWTs are rare, and account for <5% of all thoracic malignancies [10]. In our series, the majority of surgical MCWT resections were performed for breast cancer (n = 10, 23.81%), primary MCWTs (n = 9, 21.42%), and lung cancer (n = 6, 14.29%).

The postoperative 5-year survival rate in our group of patients with primary MCWTs was 87.5%, which is higher when compared to other main series, in which 5-year survival rates range from 33.3% to 68% [11,12,13,14]. This difference in survival rates may be due to the limited number of patients with primary MCWTs (n = 9, 21.42%) in our series.

The favorable 5-year overall survival rate of 51.9% and survival rate of 42.6% in 28 patients with secondary MCWTs who underwent surgery with curative intent may be explained by the heterogeneity of the tumors’ origins in our presented group.

Postoperative 5-year overall survival for the two large secondary MCWT subgroups in the analyzed series was 41.7% for patients who underwent resection for lung cancer, and 53.3% for patients who underwent resection for breast cancer, respectively. These results are comparable with other main series in which 5-year survival range from 37% to 40% for lung cancer [15,16,17] and from 43% to 58% for breast cancer [18,19,20].

We performed chest wall surgery in eight (19.04%) patients with metastatic breast cancer and in two (4.76%) patients who had direct breast cancer invasion. Some authors have suggested that there are no statistically significant differences in survival between these groups [21,22]. In our series, the 5-year survival rate was 100% for patients with direct invasion and 42.9% for patients with breast cancer metastases, respectively. We observed a large difference between these two groups, which has not been reported by other authors. This difference may be explained by the small number of patients in our series.

Surgical treatment of secondary MCWTs can be performed with curative intent, as a part of multimodal treatment, but also with a palliative intention to control pain, odor, and bleeding.

In thirteen (30.96%) cases, the MCWT resection was performed with positive margin (R1/2). Nineteen patients (45.23%), including those with positive resection margin, received postoperative adjuvant therapy. It has been suggested that incomplete (R1/R2) chest wall resection for lung cancer is a negative prognostic factor, even after adjuvant therapy, while the role of adjuvant chemotherapy in the management of primary sarcomas is still unclear [2].

The more favorable 5-year postoperative survival rate of 77.1% for incomplete (R1/R2) resection compared to 42.4% for complete (R0) resection may be due to the heterogeneity of MCWT, the limited number of patients in our series, and postoperative adjuvant therapy.

Some authors report that positive margins are the most important risk factor for the local recurrence of sarcomas, and have a significant impact on disease-free survival and overall survival [23], while other authors suggest that negative margins reduce local recurrence only in low-grade sarcomas and do not affect survival [24]. Thus, a 4 cm margin of healthy tissue determines a 56% five-year survival rate, which drops to 29% with a 2 cm safety margin [13]. For lung cancer, chest wall resection with at least a 2 cm margin of healthy tissue is recommended. Incomplete resection is a negative prognostic factor, even after adjuvant treatment [2]. It has been noted that histological type, tumor size, depth of chest wall invasion and extent of lung and chest wall resection do not correlate with survival [25]. According to other authors, deep invasion of the chest wall is associated with a worse prognosis [26].

Due to the rarity of MCWTs and their heterogeneity in the reported retrospective series, we conclude that there is no standardized safety resection margin for particular histological MCWT types.

According to some authors, biological meshes made available in recent years are resistant to infection and should not be removed in case of infection [27,28,29,30]. It has been noted that biologic meshes such as bovine pericardium (Tutomesh, Tutogen Medical, Alachua, FL, USA), porcine dermis (Strattice, LifeCell Corp., Bridgewater, NJ, USA), and the acellular dermal matrix (AlloDerm, LifeCell Corp.) have good tensile strength and elasticity, and eventually incorporate into native tissues [31]. Others report that the use of biological meshes is associated with postoperative complications such as wound infection or haematoma [32]. In our patients, in whom, according to our personal surgical experience and preferences, we used only synthetic mesh, mesh removal due to postoperative pleural empyema was necessary in two (4.76%) cases. The low rate of synthetic mesh removal after chest wall reconstruction can be explained by the elasticity, plasticity, and permeability of these meshes, as well as by surgeon’s personal experience.

We have also used titanium osteosynthesis/plate systems such as Stratos (MedXpert, Heitersheim, Germany) and the Synthes Titanium Sternal Fixation System (Synthes, Solothurn, Switzerland) for chest wall reconstruction at our institution. In particular, osteosynthesis systems are useful in multistage rib bridging and anterior chest wall stabilization [33]. It has been also noted that titanium has the ability to integrate with bone, and it is resistant to infection. Its pliability also more closely mimics physiological rib movement [34].

The current development of 3D printing technology allows implants for chest wall reconstruction to be tailored individually to the patient. This technology is often used for sternal resection in particular, and has the advantages of reduced operative time due to modern fixation systems, good aesthetic results, and reduced postoperative pain [35]. For 3D printed sternal prostheses, titanium is the most widely used material [36].

Alternatives to titanium include new bone-like materials such as polyether-ether-ketone (PEEK), which improves integration with the host with less functional impairment due to a modulus of elasticity similar to that of cortical bone [37]. However, evidence is still lacking for this material, and further studies are needed to obtain long-term data. The decision on the choice of material ultimately depends on the size of the defect and the experience of the surgeon.

Thanks to the multidisciplinary involvement of plastic surgeons specializing in microsurgery in 23 (54%) chest wall reconstructions, a variety of musculocutaneous flaps were used. In 26 (61.9%) cases, we used a muscle flap for defect coverage (prosthetic material and muscle flap—22 (52.4%), muscle flap only—4 (9.5%)).

In cases wherein pedicle/local flaps could not be used due to prior surgery or radiation therapy, free muscle flaps were an additional key to chest wall reconstruction. Fasciocutaneous or musculocutaneous flaps from the back region (latissimus dorsi flap) or from the abdominal region (transverse rectus abdominis myocutaneous flap) were some of the free flaps regularly used at our institution. The internal mammary artery and thoracodorsal vessels were the main connecting vessels on the anterior and lateral chest wall. If necessary, arteriovenous loops were created by a vascular surgeon.

Free tissue transfers have become physically less demanding surgical procedures and can now be performed with a similar or even higher degree of safety than local flap transfer as a result of the improvements in microsurgical techniques [4].

The heterogeneity of the single-center study group, limited number of patients, and retrospective nature are the main limitations of this study.

## 5. Conclusions

The interdisciplinary treatment of MCWTs involving full-thickness chest wall resection is challenging, especially for thoracic and plastic reconstructive surgeons. Nevertheless, advances in the pathology examination, a wide choice of the prosthetic materials, experience with soft-tissue coverage with muscle flap, and improvements in neoadjuvant and adjuvant therapy regimes have allowed for a significant improvement in post-operative outcomes after surgical treatment of MCWTs. With this study, we can confirm that surgical treatment of primary MCWTs is associated with acceptable long-term survival. For secondary MCWTs, surgery can be performed with both curative and palliative intent to improve quality of life, at least temporarily.

## Figures and Tables

**Figure 1 jpm-13-01405-f001:**
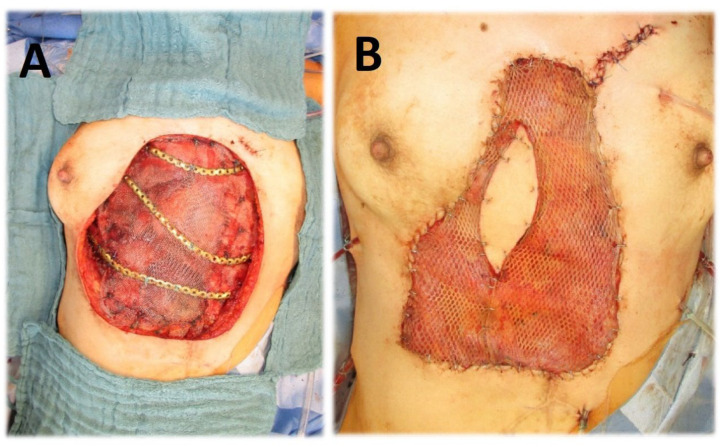
(**A**) Chest wall resection (complete sternectomy, rib resection, diaphragm resection) for necrotizing metastasis of cervical cancer; reconstruction of diaphragm with prolene-mesh; chest wall reconstruction with metal plate. (**B**) Coverage with latissimus dorsi muscular flap.

**Figure 2 jpm-13-01405-f002:**
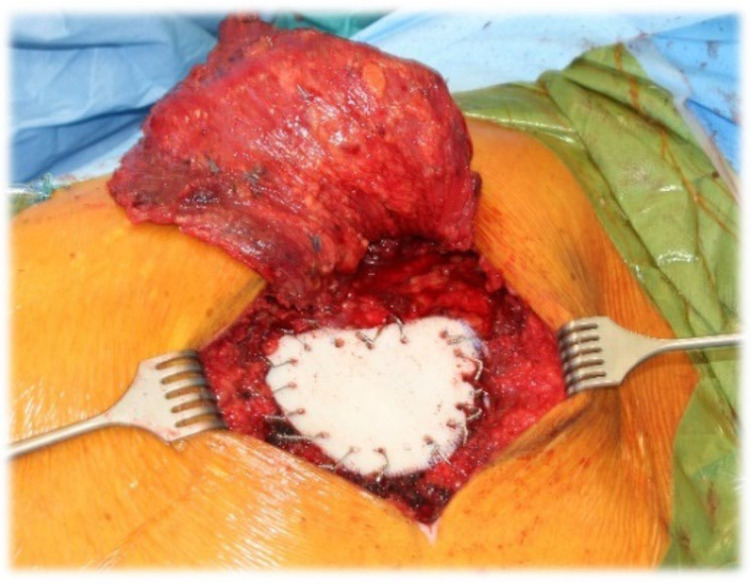
Chest wall resection (partial sternal resection, rib resection) and reconstruction using prolene mesh with bone cement, as well as coverage with a latissimus dorsi muscular flap for a chondrosarcoma of corpus sterni.

**Figure 3 jpm-13-01405-f003:**
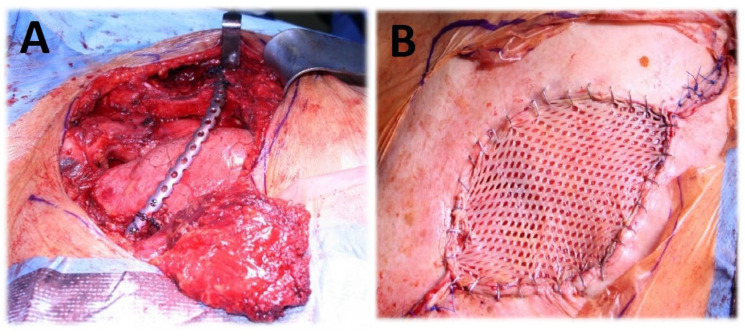
(**A**) Chest wall resection (partial claviculectomy, partial rib resection C1-3, resection of musculus pectoralis major, plate osteosynthesis, coverage with pectoralis major flap). (**B**) Split-thickness skin graft.

**Figure 4 jpm-13-01405-f004:**
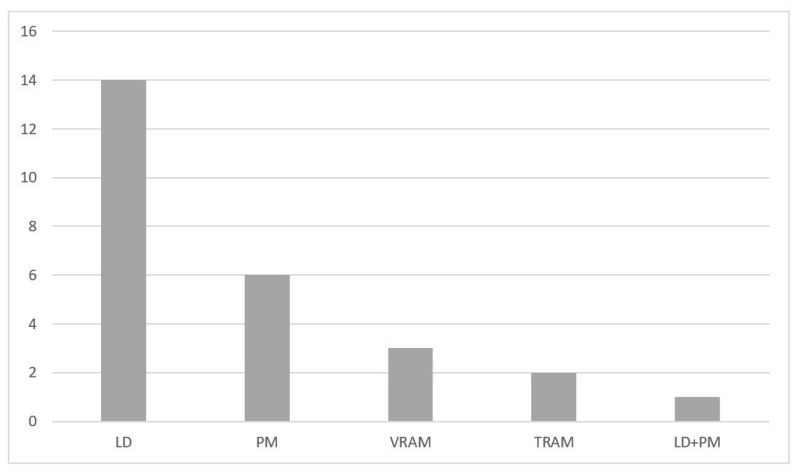
Muscle flaps used for chest wall reconstruction institution. LD, Latissimus dorsi; PM, Pectoralis major; TRAM, Transverse rectus abdominis muscle; VRAM, Vertical rectus abdominis muscle.

**Figure 5 jpm-13-01405-f005:**
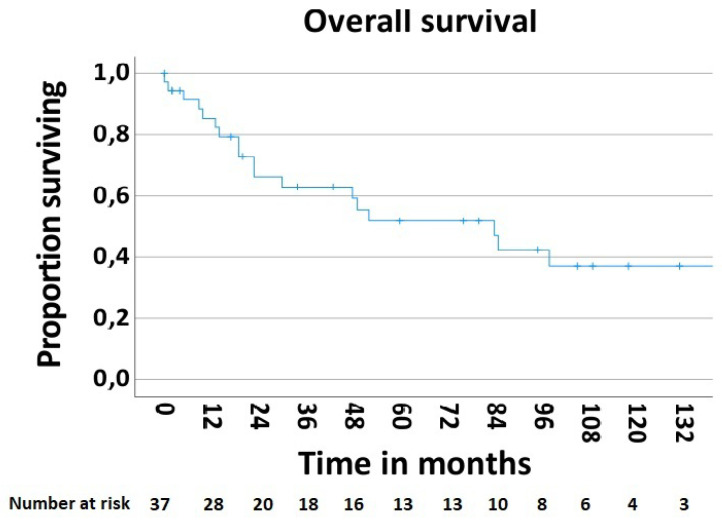
Overall survival rate after surgical treatment of MCWTs.

**Figure 6 jpm-13-01405-f006:**
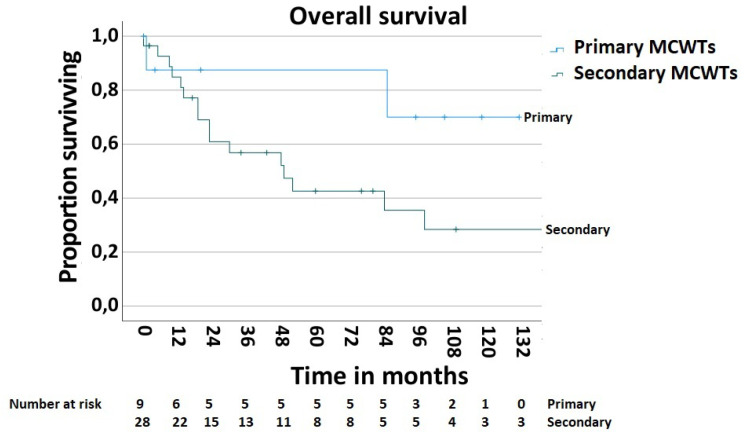
Survival after surgical treatment of MCWTs according to origin of MCWTs.

**Figure 7 jpm-13-01405-f007:**
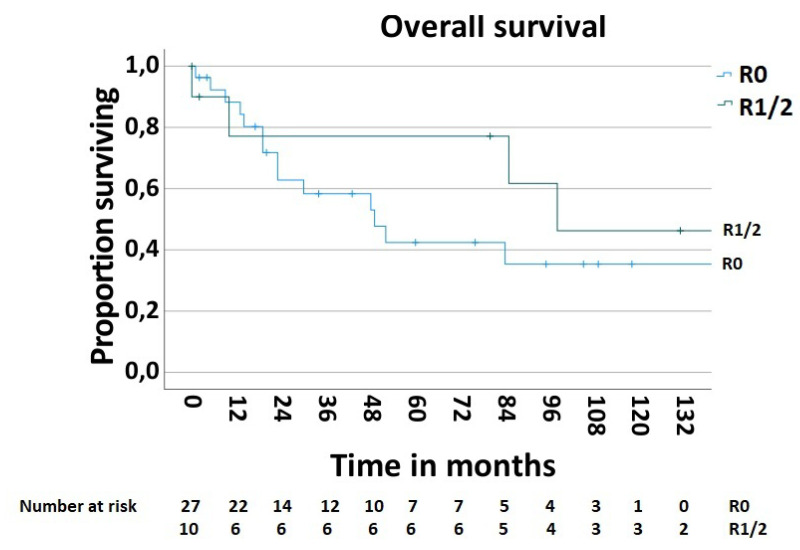
Survival after surgical treatment of MCWTs according to completeness of MCWTs resection.

**Table 1 jpm-13-01405-t001:** Characteristics of patients who underwent surgical treatment of MCWT.

Characteristics	n = 42
Male gender, n (%)	25 (59.5)
Age at surgery (y), median (range)	60 (25–86)
Secondary tumor histology, n (%)	33 (78.57)
**Histological subtypes of secondary MCWT, n (%)**	
Breast cancer	10 (23.81)
Non-small-cell lung cancer	6 (14.29)
Renal cancer	4 (9.52)
Malignant melanoma	3 (7.14)
Liposarcoma	2 (4.76)
Colorectal carcinoma	1 (2.38)
Hepatocellular carcinoma	1 (2.38)
Cholangiocellular carcinoma	1 (2.38)
Prostate carcinoma	1 (2.38)
Hemangiopericytoma	1 (2.38)
Thyroid carcinoma	1 (2.38)
Cervical cancer	1 (2.38)
Histiocytoma	1 (2.38)
**Histological subtypes of primary MCWT, n (%)**	
Chondrosarcoma	3 (7.14)
Fibromyosarcoma	1 (2.38)
Leiomyosarcoma	1 (2.38)
Liposarcoma	1 (2.38)
Fibrosarcoma	1 (2.38)
Plasmacytoma	1 (2.38)
Radiation-associated angiosarcoma after breast cancer therapy	1 (2.38)

**Table 2 jpm-13-01405-t002:** Characteristics of patients who underwent surgical treatment of MCWT with palliative intent.

Palliative intent of resection, n (%)
Ulcerating sternal malignant melanoma, 2 (4.76)
Painful rib metastasis of cholangiocellular carcinoma, 1 (2.38)
Necrotic cervical cancer metastasis, 1 (2.38)
Necrotic metastasis of cholangiocellular carcinoma, 1 (2.38)

**Table 3 jpm-13-01405-t003:** Characteristics of neoadjuvant therapy according to the origin of the MCWTs.

	Primary MCWT, n (%)	Secondary MCWT, n (%)
Radio-chemotherapy	Chondrosarcoma, n = 1 (2.38) Plasmacytoma, n = 1 (2.38) Leiomyosarcoma, n = 1 (2.38)	Malignant melanoma, n = 2 (4.76) Non-small-cell lung cancer, n = 2 (4.76) Histiocytoma, n = 1, (2.38) Cervical cancer, n = 1 (2.38)
Chemotherapy		Hemangiopericytoma, n = 1 (2.38)
Radiotherapy		Cholangiocellular carcinoma, n = 1 (2.38)

**Table 4 jpm-13-01405-t004:** Postoperative complications after chest wall reconstruction according to the Clavien–Dindo classification.

Grades	Number of Patients	Complications (Treatment)
I	2	Wound seroma (conservative management with Redon drain) Recurrent laryngeal nerve paresis (conservative management)
II	1	Wound seroma (conservative management) Left subclavian vein thrombosis (oral anticoagulation)
III/IV	9	Acute bleeding (surgical revision) n = 1 Pleural effusion (chest drain) n = 1 Pleural empyema (surgical revision) n = 2 Wound hematoma (surgical revision) n = 1 Wound dehiscence (surgical revision) n = 1 Hemothorax (surgical revision) n = 3
V	1	Death

**Table 5 jpm-13-01405-t005:** Muscle flaps used for chest wall reconstruction according to location of the resection.

Muscle Flap	Number (%)	Partial Sternectomy	Complete Sternectomy	Chest Wall Resection Anterior	Chest Wall Resection Antelateral	Chest Wall Resection Lateral
Latissimus dorsi (LD)	14 (33.3)	5	1	2	2	4
Pectoralis major (PM)	6 (14.3)	4	0	1	0	1
Vertical rectus abdominis (VRAM)	3 (7.14)	1	0	0	0	2
Transverse rectus abdominis (TRAM)	2 (4.76)	1	0	0	0	1
Latissimus dorsi (LD) + Pectoralis major (PM)	1 (2.38)	0	0	1	0	0

**Table 6 jpm-13-01405-t006:** Prosthetic materials used for chest wall reconstruction.

Prosthetic Material	Product Name	Number (%)
Polypropylene	Prolene-Mesh (Ethicon, Inc., Somerville, NJ, USA)	17 (40.4)
Polyethylene and polypropylene	Marlex-Mesh (Davol & Bard, Warwick, RI, USA)	7 (16.7)
Polyglactin	Vicryl-Mesh (Ethicon, Inc., Somerville, NJ, USA)	5 (11.9)
Polyester with collagen film	Parietex-Mesh (COVIDIEN, Medtronic, Dublin, Irland)	2 (4.76)
Polyester	Polyester (Dacron^®^, DuPont de Nemours, Inc., Wilmington, DE, USA) Mesh	1 (2.38)
Polytetrafluoroethylene (PTFE)	Gore-Tex (W. L. Gore & Associates, Newark, DE, USA)	1 (2.38)

**Table 7 jpm-13-01405-t007:** Characteristics of the adjuvant therapy according to the origin of the MCWTs and completeness of the resection of MCWTs.

Adjuvant Therapy	Primary MCWT, n (%), R	Secondary MCWT, n (%), R
Radio-chemotherapy, n = 8 (19.04)	Chondrosarcoma, n = 1 (4.76), R1/2 Liposarcoma, n = 1 (2.38), R1/2	Breast cancer, n = 2 (4.76), R0 Cholangiocellular carcinoma, n = 1 (2.38), R1/2 Renal cancer, n = 1 (2.38), R0 Thyroid carcinoma, n = 1 (2.38), R1/2 Malignant melanoma, n = 1 (2.38), R1/2
Chemotherapy, n = 7 (16.6)		Malignant melanoma, n = 1 (2.38), R1/2 Histiocytoma, n = 1 (2.38), R0 Liposarcoma, n = 1 (2.38), R0 Non-Small Cell Lung Cancer, n = 1 (2.38), R0 Renal cancer, n = 1 (2.38), R0 Breast cancer, n = 1 (2.38), R1/2 Hemangiopericytoma, n = 1 (2.38), R0
Radiotherapy, n = 3 (7.14)	Chondrosarcoma, n = 1 (2.38), R1/2	Breast cancer, n = 2 (4.76), R1/2
Hormone therapy, n = 1 (2.38)		Breast cancer, n = 1 (2.38), R0

## Data Availability

Due to the nature of the medical records analyzed here, the underlying data cannot be publicly shared.

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
