# Peer review of "Interdisciplinary Treatment of Malignant Chest Wall Tumors"

_jpm, 2023, doi:10.3390/jpm13091405_

Round 1
Reviewer 1 Report
Thank you for the opportunity to analyze your article.
In this article, authors have reviewed their experience of chest wall resection and reconstruction for primary and metastatic malignant chest wall tumors (MCWTs), with a focus on perioperative outcomes for short term events and overall survival (OS) for long term events.
This is an interesting topic dealing with aggressive tumors, where we have to balance oncological concerns, functional concerns and quality of life of the patient.
Concerning the introduction:
The introduction is well written, highlighted main oncological priority depending on the histology of the MCWT.
Depending on the histology, the treatment plan, about the kind of resection, its extension and the king of reconstruction can be established with different surgeons.
Quality of life is one of the major concerns for palliative indications. For some specific cases, we can offer with surgery, a better quality of life, but we need to select patients.
Concerning the methodology:
No major concerns except:
- Your target resection margin can’t always be 2 to 4 cm. It will depend on the histology of the MCWT. Need to bring margins adapted
- Depending on the location of the resection, a scheme can be interesting for muscle flaps you used to apply.
Concerning the results
Results are well reported and clearly presented, but a table with palliative indications, peri operative treatment can be interesting.
Post operative complications are well reported using the right classification.
For the extent of the resection a table will be more informative, including the completeness of the resection, and its extent. Associated with the histology, can you give the size of your margins for each histology in mean?
Concerning the prosthetic materials you used, do you have some data or it’s more according surgeons’ habits.
Concerning the long-term outcomes, the histology need to be taken in account more than primary and secondary MCWT. It’s difficult to understand why the 5-year OS is longer for R1/R2 population compared to R0 group, but it may be impacted by the size of your population/ Or by the adjuvant Radiotherapy administrated?
Concerning the discussion:
It’s a well written discussion with good references highlighted your very good results by the size of your population, with also a selection bias. Your results are interesting according long-term OS, with also good short term outcomes. As you have reported, the heterogeneity of the population and “it’s small” but still impressive size may have played a role on your good results.
Concerning the discussion dealing with the reconstruction, it seems to be too short. Maybe we would appreciate a “more extensive review” about the kind of meshes, the type or osteosynthesis devices that can be used, the role of the different muscle’s flaps also.
As you have briefly mention, today our surgery is imaging guided. We have 3D reconstruction, 3D models, IA surgery assisted, which are helpful for muscle flaps to identify and locate the arising vessels. Maybe you can bring a more detailed review.
Concerning the conclusion:
Your results are very interesting, and your limitations are well reported.
But we would appreciate a longer and more detailed discussion about the reconstruction dealing with the multidisciplinary approach, advantages of pre operative imaging, advantages of the different osteosynthesis devices, meshes and muscle flaps.
It’s a well written, easy reading and interesting article, that need some precisions.
Congratulations to authors for this work.
Author Response
Dear Editor,
I am pleased to submit a revised version of our manuscript entitled “Interdisciplinary Treatment of Malignant Chest Wall Tumors”.
The revision was performed in accordance with the suggestions of the reviewers and Editorial Office.
All changes performed have been highlighted in green.
Please note the increased (n =38) number of the references.
With regards to the word count, according to my software, it is now close to 4,000. Please note that this original paper was written by the surgeons, who prefer very pregnant writing style. We believe that adding more theoretical aspects to the text of the manuscript would only dilute the clue of the study and turn it to a thesis rather than compact surgical article.
However, if you do not agree with our point of view, please do not hesitate to contact me and I will be more than happy to add some more text.
I appreciate your excellent assistance in the submission process and hope that this revised version of our manuscript meets the criteria for publication in your prestigious Journal of Personalised Medicine.
Your Sincerely
Review 1.
- - Your target resection margin can’t always be 2 to 4 cm. It will depend on the histology
of the MCWT. Need to bring margins adapted:
„ Complete (R0) tumor resection was a key surgical principle. However, the intraoperative finding of more advanced tumor invasion compared to the carefully performed preoperative imaging was not an indication to discontinue surgery. Therefore, a specific safety margin for resection has not been defined.”
- - Depending on the location of the resection, a scheme can be interesting for muscle flaps you used to apply.
“Table 5. Muscle flaps used for chest wall reconstruction according to location of the resection”
- Results are well reported and clearly presented, but a table with palliative indications, peri operative treatment can be interesting.
“Table 2. Characteristics of patients who underwent surgical treatment of MCWT with palliative intent.”
- Concerning the prosthetic materials you used, do you have some data or it’s more according surgeons’ habits.
“In our patients, in whom, according to our personal surgical experience and preferences, we used only synthetic mesh, the mesh removal due to postoperative pleural empyema was necessary in 2 (4.76%) cases.”
- Concerning the long-term outcomes, the histology need to be taken in account more than primary and secondary MCWT. It’s difficult to understand why the 5-year OS is longer for R1/R2 population compared to R0 group, but it may be impacted by the size of your population/ Or by the adjuvant Radiotherapy administrated?
“The more favorable 5-year postoperative survival rate of 77.1% for incomplete (R1/R2) resection compared to 42.4% for complete (R0) resection may be due to the heterogeneity of MCWT, the limited number of patients in our series as well as postoperative adjuvant therapy.”
- Concerning the discussion dealing with the reconstruction, it seems to be too short. Maybe we would appreciate a “more extensive review” about the kind of meshes, the type or osteosynthesis devices that can be used, the role of the different muscle’s flaps also. As you have briefly mention, today our surgery is imaging guided. We have 3D reconstruction, 3D models, IA surgery assisted, which are helpful for muscle flaps to identify and locate the arising vessels. Maybe you can bring a more detailed review.
“According to some authors, available in recent years biological meshes are resistant to infection and should not be removed in case of infection [28][29][30][31]. It has been noted, that biologic meshes, such as bovine pericardium (Tutomesh, Tutogen Medical, Alachua, FL), porcine dermis (Strattice, LifeCell Corp, Bridgewater, NJ), and acellular dermal matrix (AlloDerm, LifeCell Corp) have good tensile strength and elasticity and eventually incorporate into native tissues [32] . Others report that the use of biological meshes is associated with postoperative complications such as wound infection or haematoma [33]. In our patients, in whom, according to our personal surgical experience and preferences, we used only synthetic mesh, the mesh removal due to postoperative pleural empyema was necessary in 2 (4.76%) cases. The low rate of synthetic mesh removal after chest wall reconstruction can be explained by the elasticity, plasticity, permeability of these meshes as well as by surgeon's personal experience.
We have also used titanium osteosynthesis/plate systems as Stratos (MedXpert, Heitersheim, Germany) and Synthes Titanium Sternal Fixation System (Synthes, Solothurn, Switzerland) for chest wall reconstruction at our institution. Especially, the osteosynthesis systems are useful in multistage rib bridging or anterior chest wall stabilization [34]. It has been also noted, that titanium has the ability to integrate with bone, and it is resistant to infection. Its pliability also more closely mimics physiologic rib movement [35].
The current development of 3D printing technology allows tailoring the implants for chest wall reconstruction individually to the patient. This technology is often used for sternal resection in particular, and has the advantages of reduced operative time due to modern fixation systems, good aesthetic results and reduced postoperative pain [36]. For 3D printed sternal prostheses, titanium is the most widely used material [37].
Alternatives to titanium include new bone-like materials such as polyether-ether-ketone (PEEK), which improves integration with the host with less functional impairment due to a modulus of elasticity similar to that of cortical bone [38]. However, evidence is still lacking for this material and further studies are needed to obtain long-term data. The decision on the choice of material ultimately depends on the size of the defect and the experience of the surgeon.”
Reviewer 2 Report
I read with interest this article concerning chest wall resections for malignant chest wall tumors. There are obvious limitations but these are clearly admitted: heterogeneity of the study group, limited number of patients and retrospective nature. There's a lackness of originalty.
Some revisions are required
Pay attention to the punctuation (for example sentence 233-235)
I didn't understand sentence 271-273
Author Response
Dear Editor,
I am pleased to submit a revised version of our manuscript entitled “Interdisciplinary Treatment of Malignant Chest Wall Tumors”.
The revision was performed in accordance with the suggestions of the reviewers and Editorial Office.
All changes performed have been highlighted in green.
Please note the increased (n =38) number of the references.
With regards to the word count, according to my software, it is now close to 4,000. Please note that this original paper was written by the surgeons, who prefer very pregnant writing style. We believe that adding more theoretical aspects to the text of the manuscript would only dilute the clue of the study and turn it to a thesis rather than compact surgical article.
However, if you do not agree with our point of view, please do not hesitate to contact me and I will be more than happy to add some more text.
I appreciate your excellent assistance in the submission process and hope that this revised version of our manuscript meets the criteria for publication in your prestigious Journal of Personalised Medicine.
Your Sincerely
Review 2.
- Pay attention to the punctuation (for example sentence 233-235)
“Postoperative 5-year overall survival for the two large secondary MCWT subgroups in the analyzed series was 41.7% for patients, who underwent resection for lung cancer and 53.3% for breast cancer, respectively.”
- I didn't understand sentence 271-273
“According to some authors, available in recent years biological meshes are resistant to infection and should not be removed in case of infection [28][29][30][31].”

Reviewer 3 Report
This study looks to review the perioperative outcomes associated with chest wall resection for malignant chest wall tumours. The authors report on their experience of performing these procedures and include good coverage of the issues that can arise with such procedures.
Whilst these details are good, I think given the relatively small number of patients they report on, and the variety seen in terms of parameters such as source tumour, age range, complications seen and prosthetic materials used, the ability to make concrete, statistically significant conclusions from the study are quite weak, and I think the paper could be strengthened by expanding the study to include other centres, or by incorporating reported data from the literature to add weight to the analyses.
Other comments:
The introduction seems light, and would be improved by more extensive detail on the options for the reconstructive procedures used, and on the complications arising from this surgery.
More detail of the statistical analyses used should be given in the methodology.
Standard of English is very good overall, with just a few errors.
Author Response
Dear Editor,
I am pleased to submit a revised version of our manuscript entitled “Interdisciplinary Treatment of Malignant Chest Wall Tumors”.
The revision was performed in accordance with the suggestions of the reviewers and Editorial Office.
All changes performed have been highlighted in green.
Please note the increased (n =38) number of the references.
With regards to the word count, according to my software, it is now close to 4,000. Please note that this original paper was written by the surgeons, who prefer very pregnant writing style. We believe that adding more theoretical aspects to the text of the manuscript would only dilute the clue of the study and turn it to a thesis rather than compact surgical article.
However, if you do not agree with our point of view, please do not hesitate to contact me and I will be more than happy to add some more text.
I appreciate your excellent assistance in the submission process and hope that this revised version of our manuscript meets the criteria for publication in your prestigious Journal of Personalised Medicine.
Your Sincerely
Review 3.
- Whilst these details are good, I think given the relatively small number of patients they report on, and the variety seen in terms of parameters such as source tumour, age range, complications seen and prosthetic materials used, the ability to make concrete, statistically significant conclusions from the study are quite weak, and I think the paper could be strengthened by expanding the study to include other centres, or by incorporating reported data from the literature to add weight to the analyses.
“The postoperative 5-year survival rate in our group of patients with primary MCWTs was 87.5%, which is higher when compared to other main series with 5-year survival rates ranging from 33.3% to 68% [6-9].”
“Postoperative 5-year overall survival for the two large secondary MCWT subgroups in the analyzed series was 41.7% for patients, who underwent resection for lung cancer and 53.3% for breast cancer, respectively. These results are comparable with other main series with 5-year survival ranging from 37% to 40 % for lung cancer [15][16][17] and from 43% to 58% for breast cancer [18][19][20].”
“We performed chest wall surgery in 8 (19.04%) patients with metastatic breast cancer and in 2 (4.76%) patients who had direct breast cancer invasion. Some authors have suggested that there are no statistically significant differences in survival between these groups [21][22]. At our series, the 5-year survival rate was 100 % for patients with direct invasion and 42.9% for patients with breast cancer metastases respectively. We observed a large difference between these two groups, which has not been reported by other authors. This difference can be explained by the small number of patients in our series.”
- The introduction seems light, and would be improved by more extensive detail on the options for the reconstructive procedures used, and on the complications arising from this surgery.
“According to some authors, available in recent years biological meshes are resistant to infection and should not be removed in case of infection [28][29][30][31]. It has been noted, that biologic meshes, such as bovine pericardium (Tutomesh, Tutogen Medical, Alachua, FL), porcine dermis (Strattice, LifeCell Corp, Bridgewater, NJ), and acellular dermal matrix (AlloDerm, LifeCell Corp) have good tensile strength and elasticity and eventually incorporate into native tissues [32] . Others report that the use of biological meshes is associated with postoperative complications such as wound infection or haematoma [33]. In our patients, in whom, according to our personal surgical experience and preferences, we used only synthetic mesh, the mesh removal due to postoperative pleural empyema was necessary in 2 (4.76%) cases. The low rate of synthetic mesh removal after chest wall reconstruction can be explained by the elasticity, plasticity, permeability of these meshes as well as by surgeon's personal experience.
We have also used titanium osteosynthesis/plate systems as Stratos (MedXpert, Heitersheim, Germany) and Synthes Titanium Sternal Fixation System (Synthes, Solothurn, Switzerland) for chest wall reconstruction at our institution. Especially, the osteosynthesis systems are useful in multistage rib bridging or anterior chest wall stabilization [34]. It has been also noted, that titanium has the ability to integrate with bone, and it is resistant to infection. Its pliability also more closely mimics physiologic rib movement [35].
The current development of 3D printing technology allows tailoring the implants for chest wall reconstruction individually to the patient. This technology is often used for sternal resection in particular, and has the advantages of reduced operative time due to modern fixation systems, good aesthetic results and reduced postoperative pain [36]. For 3D printed sternal prostheses, titanium is the most widely used material [37].
Alternatives to titanium include new bone-like materials such as polyether-ether-ketone (PEEK), which improves integration with the host with less functional impairment due to a modulus of elasticity similar to that of cortical bone [38]. However, evidence is still lacking for this material and further studies are needed to obtain long-term data. The decision on the choice of material ultimately depends on the size of the defect and the experience of the surgeon.”
- More detail of the statistical analyses used should be given in the methodology.
“The standardized Clavien-Dindo classification was used to estimate postoperative complications [5]. Postoperative survival was calculated using Kaplan-Meier survival analysis.”
